# Open Innovation with Relational Capital, Technological Innovation Capital, and International Performance in SMEs

**Dongwoo Ryu [1], Kwang Ho Baek [2] and Junghyun Yoon [1,\*]**

1   School of Business, Yeungnam University, Gyeongsan 38541, Korea; rdw999@ynu.ac.kr
2   Korea Employment Information Service, Jincheon 27740, Chungcheongbuk-do, Korea; gorams69@keis.or.kr
\*   Correspondence: jyoon@yu.ac.kr; Tel.: +82-10-9385-7401

**Abstract:** The importance of international markets is constantly emphasized for small and medium enterprises(SMEs). In previous studies, technological innovation capabilities were emphasized as a factor that enables SMEs to compete in the international market. To this end, SMEs need to cooperate with external partners to strengthen their technological innovation capabilities to thus improve their international performance. With the perspective view of open innovation, this research explores the effects of relational capital and technological innovation capability on international performance, with a particular focus on the moderating effect of alliance proactiveness. Building on previous literature regarding internationalization, technological innovation, and alliance proactiveness, research hypotheses were developed and tested using data collected from 175 SMEs. A hierarchical regression analysis was applied. The analysis showed that, first, relational capital had a significant effect on the technological innovation capability. Second, technological innovation capability has a significant influence on the international performance. Third, technological innovation capability mediated the relationship between relational capital and international performance. Finally, alliance proactiveness was found to moderate the relationship between technological innovation capability and international performance. The key research findings imply that relational capital and alliance proactiveness are the key factors of international performance, as they improved the development of the technological innovation capability.

**Keywords:** relational capital; technological innovation capability; alliance proactiveness; international performance

## 1. Introduction

Recently, the importance of advancing into international markets is constantly being emphasized for small and medium enterprises(SMEs). Firms entering international markets have various economic advantages. That is, as the market size increases, the market power is strengthened according to the effects of economies of scale and scope, and this is a very important activity for firms because it helps to increase productivity and profitability [1]. Previous studies emphasized the importance of strategic assets, such as technological innovation capabilities, as a factor that enables SMEs to compete in international markets [2]. Therefore, the importance of firms with high technological innovation capabilities is increasing even more because they have the advantage of being able to maintain competitiveness through differentiation in international markets [3].

Firms are demanding openness of their innovation processes more than ever before [4]. Open innovation can increase the possibility of creating innovation by combining the knowledge accumulated internally through the search for external knowledge for technological innovation and the knowledge possessed by external actors [5–7]. Despite this importance, open innovation studies have been mainly focused on large firms, and studies on SMEs are insufficient [8,9]. In particular, research on how SMEs use external sources of knowledge for open innovation is insufficient [10,11]. Accordingly, we focused on the open innovation process of SMEs.

Sustainability has been considered a major means of securing a competitive advantage for SMEs [12,13]. This is because sustainability can enable the introduction of new products, production processes, management practices, or business methods with economic, social, and environmental consequences [14,15]. Research on sustainability and open innovation argues that collaboration with external partners helps firms to improve their innovation performance [16]. SMEs will need to pay attention to relational capital that secures knowledge resources through cooperation with external stakeholders and helps build technological innovation capabilities.

Relational capital is attracting attention as a factor that influences the innovation performance of SMEs as it uses external knowledge and experience for technological innovation [6]. Stakeholders, such as customers, suppliers, and competitors, help SMEs to provide various sources of knowledge [10]. SMEs can combine external knowledge with their existing experience and internal knowledge to help solve problems related to products and processes. Through this, it will be possible to build technological innovation capabilities through the creation of new ideas [17].

In addition, in the current competitive environment where the importance of open innovation is emphasized, the importance of the networking role of external knowledge exploration is increasing. In particular, to strengthen the technological innovation capabilities of SMEs, it is important to expand the networks related to internationalization. This means that the role of strategic alliances that can compensate for scarce resources for SMEs is very important. Strategic alliances can help overcome weaknesses in internationalization by supplementing strategic assets through acquiring knowledge and information about international markets [18]. Therefore, the importance of alliance proactiveness to find strategic alliance opportunities with potentially valuable new firms in an uncertain foreign market environment is further emphasized [6,10].

To fill this gap in the literature, the purpose of this study was to verify the causal relationship between relational capital, technological innovation capability, and international performance, and to examine the moderating effect of alliance proactiveness in these relationships. The research results are expected to contribute to increasing the importance of alliance proactiveness in enhancing the international performance. The remainder of the paper is arranged as follows. The literature review and hypotheses development are presented in Section 2. Then, research methods and empirical results are presented in Sections 3 and 4, respectively. Finally, contributions, implications, limitations, and suggestions for future research are discussed in Section 5.

## 2. Literature Review and Hypotheses

### 2.1. Open Innovation Perspective

Open innovation is defined as accelerating internal innovation and maximizing value by appropriately leveraging the flow of knowledge into and out of the enterprise, and by expanding the market for the external use of innovation [19,20]. This type of innovation is characterized by different innovators, the formation of multi-organizational relationships among them, and internal and external paths to access distributed knowledge [21]. Open innovation activities of the firm are known to reduce innovation costs (R&D) and shorten the time it takes to commercialize new ideas as well as create new sources of revenue, positively impacting the innovation productivity [19,22]. In previous studies, open innovation was based on continuous interaction and openness with other organizations, suggesting that cooperation with external networks promotes innovation activities and reduces innovation costs and risks, thus improving innovation performance [23].

The key to open innovation is in how firms use the ideas and knowledge of external partners in the innovation process [6]. Firms explore various external entities to acquire the knowledge required for innovation activities, and each entity has different characteristics in terms of the type of knowledge and accessibility it possesses. In general, firms collaborate with universities and research institutes to acquire knowledge [24] as well as suppliers, customers, and competitors [25].

Recently, the scope has expanded to realize open innovation through cooperation with foreign firms [26]. The knowledge that the firm wants to understand from the outside is not only limited to "new technical knowledge" but also contains "extant market knowledge" linked to production, marketing, and customer knowledge. In this study, we focused on exploratory search and exploitative search [27].

In this regard, in early research on knowledge, search activities were classified as local search and non-local search in the horizontal dimension of knowledge search scopes [28]. The former contributes to incremental innovation through internal exploitative search activities within the scope of the existing knowledge that is possessed. The latter, on the other hand, contributes to radical innovation and external exploratory search activity on the new knowledge domain through the extension of the search domain. This is followed by attempts to identify exploratory and exploitative knowledge, arguing that the types and impacts of enterprise search activities differ not only by horizontal dimension, but also by vertical dimension [19].

### 2.2. Relational Capital and Technological Innovation Capability

Intellectual capital is defined as the set of intangible assets that a company requires so as to achieve its competitive advantage [29]. Previous studies argued that intellectual capital consists of human capital, structural capital, and relational capital [30,31]. First, human capital refers to the capabilities and skills of the employees within an organization. Second, structural capital can add value to an organization, such as databases, organization charts, process manuals, strategies, and customs, and indicates the knowledge of non-human factors stored in the organization. Third, relational capital refers to intangible assets that can be acquired through relationships between external companies and customers [32]. Among these capitals, relational capital can be said to be crucial because it contains the tacit knowledge inherent in the value chain [33].

Relational capital is defined as the ability to absorb and utilize necessary knowledge through the search for external knowledge in the relationships of a firm's value chain [34]. This includes all the knowledge inherent in relationships with stakeholders, such as customers, suppliers, and competitors [29]. Relational capital not only improves access to knowledge sources because it creates trust in relationships with partners but also plays a role in facilitating the exchange of knowledge by increasing expectations and motivation for the value of knowledge [35,36].

Relational capital has the following advantages. First, the customer provides the knowledge to better reflect the market's requirements for the product, service, or process for innovation. Second, suppliers can provide information on quality improvement and cost reduction. Third, technical cooperation with competitors can be complex and dangerous; however, if suppliers can identify common goals, the possibility of utilizing technology development through external resources can be significantly increased [37,38]. Due to these characteristics, firms inevitably focus on their relationships with customers, suppliers, and competitors [39].

From the perspective of open innovation, researchers emphasized that it is important for firms to acquire and utilize external knowledge beyond what they possess internally [19]. From this point of view, relational capital can be developed as a factor that causes integration between trading partners because it is combined into common knowledge by providing opportunities for knowledge exchange through mutual exchange and communication between firms and through information sharing activities [40,41]. Accordingly, firms can increase the possibility of building technological innovation capabilities by combining the knowledge accumulated inside and various knowledge possessed by external actors [5–7].

Relational capital is an essential factor in building the technological innovation capabilities of firms because it promotes integration and dissemination as well as the acquisition of external knowledge [42]. Saxenian [43] argued that knowledge accumulation formed through long-term cooperative relationships can efficiently combine and reorganize a firm's

owned resources to build innovation capabilities. Therefore, the relational capital of SMEs can build up the necessary technological innovation capabilities in the international market because it is possible to accumulate knowledge by improving the quality of relationships based on communication and trust with partners. Based on the preceding discussions, the following hypothesis is proposed:

**Hypothesis 1 (H1).** *Relational capital has a positive effect on technological innovation capability.*

### 2.3. Technological Innovation Capability and International Performance

Technological innovation capability is defined as the ability to accept and combine various types of technological knowledge and resources by executing a firm's competitive strategy and creating value [44]. Technological innovation competence is the acquisition of ideas or components to reinforce complementary knowledge and transform it into an economic advantage, so that it can preempt the speed of the commercialization of technology and market scope, which has a positive effect on the firm performance [45]. Therefore, technological innovation capability can be said to be an essential factor that can enhance a competitive advantage by planning products and technologies necessary to meet customer needs and shortening the development period [46].

According to the resource-based perspective, the technical competence of a firm is discussed as a major factor in determining the international tendency or internationalization of a firm [47]. Among the intangible resources of a firm, the technological innovation capability, which can be referred to as a technical resource, can help to advance into international markets as it affects the improvement of the competitiveness of a firm [48,49]. This is because it is possible to promote internationalization based on differentiated technologies due to the characteristics of SMEs with insufficient resources and to elicit positive responses from customer firms to these technologies.

According to Coombs and Bierly [50], technological innovation capabilities resulting from innovation activities are the determinants of increasing international performance. In addition, Zahra and Nielsen [45] argued that SMEs can not only promote successful internationalization but also create more than a certain level of performance if they possess technological innovation capabilities that their competitors cannot imitate. Based on the preceding discussions, the following hypothesis is proposed:

**Hypothesis 2 (H2).** *Technological innovation capability has a positive effect on international performance.*

### 2.4. The Mediating Effect of Technological Innovation Capability

Technological innovation capability can be said to be an important factor in gaining a competitive advantage through success in international markets. Many firms consider this capability to be a factor that increases international performance through technological innovation capabilities. In particular, technological innovation capability is an important factor in securing the core competencies and competitive advantages of a firm and has been emphasized as a mediating effect in various studies [51]. SMEs can reinforce their technological innovation capabilities by securing knowledge information through relational capital. This is because new knowledge and information can be acquired, and innovative results can be derived through the fusion of novel knowledge [52].

Relational capital has the advantage of being able to accompany interactions by promoting efficient functioning with firms in the value chain relationship held by firms. Therefore, it is possible to create new knowledge, information, and differentiated ideas in connection with customers, suppliers, and competitors, thereby acquiring new knowledge and building innovative capabilities [53].

The technological innovation capabilities, thus, form not only develop unique products based on technological superiority but also form differentiated strategic advantages. This is because firms with high technological innovation capabilities provide opportunities

to acquire a potential monopoly market by pioneering new markets, and enable cost reductions to remain competitive in international markets [3]. Dhanaraj and Beamish [54] argued that SMEs with unique technological innovation capabilities can improve international performance through economies of scale and economies of scope.

As a result, we expect that the relational capital possessed by a firm will build high-level technological innovation capabilities. We assume that the technological innovation capabilities formed through this can lead to an improvement in the international performance through the development of new technologies and products that can identify the environment required for international markets and adapt to changes. Based on the discussions, the following hypothesis is proposed:

**Hypothesis 3 (H3).** *The relationship between relational capital and international performance is mediated by their technological innovation capability.*

### 2.5. The Moderating Effect of Alliance Proactiveness

Alliance proactiveness is defined as the propensity to find strategic alliance opportunities with potentially valuable firms [55]. This tendency helps firms to develop relationships with new partners through newly formed alliances. As firms have the advantage of being able to preoccupy competitors through alliance proactiveness [56], they can conclude strategic alliances ahead of competitors or better predict the outcome of alliances, leading to the success of alliances [57]. Therefore, alliance proactiveness can be a major means of creating an environment favorable for alliances between firms and securing a competitive advantage by securing resources [58].

From the dynamic capability point of view, researchers argued that the differences in performance between firms are dependent on their ability to sense, seize, and reconfigure knowledge outside the firm [59]. A firm's proactiveness provides a strategic posture for external environmental information [60]. This is an important factor because it helps firms to understand the environment and discover the market requirements and opportunities for new resource acquisition. From this perspective, alliance proactiveness can be said to be an important basis for a firm's daily sensing activities as it helps firms seize opportunities to acquire resources that can respond to the market demands [61].

Firms with high alliance proactiveness can conduct strategic alliances in homogeneous and heterogeneous fields to cope with uncertainties in international markets. That is, based on the skills or capabilities of partners, collaborative processes, such as technology transfer and joint research, can provide a foundation for expanding the retention of knowledge [62,63]. The explorative and exploitative knowledge introduced through this is combined with the prior knowledge in the firm, embodied, transformed, and used, and thus it will be possible to reinforce the firm's technological innovation capabilities [64]. Based on these arguments, when alliance proactiveness is high, SMEs will be able to build strategic alliance opportunities and build technical knowledge through collaboration with new partners, thereby enhancing their technological innovation capabilities. Based on the discussions, the following hypothesis is proposed:

**Hypothesis 4 (H4).** *The relationship between technological innovation capability and international performance is moderated by alliance proactiveness.*

### 2.6. Research Model

Figure 1 illustrates the research model, which controls for variables, including firm age, firm, firm size, industries, alliance experience and degree of internationalization.

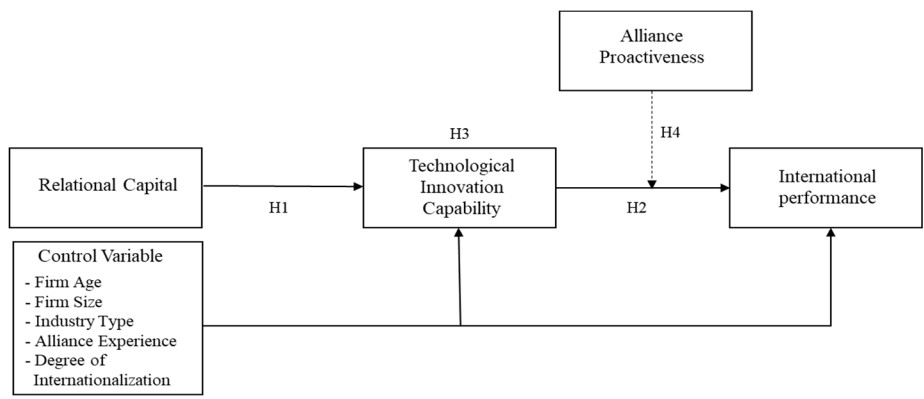

**Figure 1.** Research model.

## 3. Methods

### 3.1. Sample and Data Collection

This study focused on the manufacturing industry in South Korea. The sample included 400 SMEs among those registered in the Korea Chamber of Commerce and Industry (KCCI) database. The selection criteria were as follows. First, according to the Framework Act on SMEs, firms with a three-year average sales of more than 150 billion won were excluded. Second, firms that did not enter overseas markets or whose performance was missing were also excluded. Finally, companies with less than 50 employees were also excluded from the scope, considering the scale that could smoothly promote strategic alliances.

To collect the data, a questionnaire was developed based on previous studies of relational capital, technological innovation capability, alliance proactiveness, and the related variables. Data collection was conducted from 16 August to 2 September 2016 with the help of specialized research institutes. To increase the response rate, we made as many direct visits as possible so that the directors of the alliance or R&D departments could respond directly. To ensure data accuracy, the CEOs and managing directors who were most knowledgeable about their strategic alliances were asked to complete the questionnaire. A total of 400 questionnaires were sent out and 204 responses were received, yielding a response rate of 51.0%. Of the 204 responses, 29 were excluded from the analysis due to missing data and insufficient R&D alliance experience. As a result, a total of 175 responses were obtained for the final analysis. The description of the sample characteristics is shown in Table 1.

**Table 1.** Sample characteristics (*n* = 175).

| Characteristic | | Frequency | % |
|---|---|---|---|
| Year of establishment | Before 1990 | 50 | 28.6 |
| | 1991–2000 | 55 | 31.4 |
| | 2001–2010 | 59 | 33.7 |
| | 2011 After | 11 | 6.3 |
| Number of employees | 50–100 | 81 | 46.3 |
| | 101–200 | 50 | 28.6 |
| | 201–300 | 26 | 14.9 |
| | 301–500 | 18 | 10.3 |
| Industry type | Textile | 19 | 10.9 |
| | Machinery | 101 | 57.7 |
| | Electronics | 36 | 20.6 |
| | Other | 19 | 10.9 |

### 3.2. Measures

The variables used in this study were measured as follows. First, relational capital was defined as the ability to absorb and utilize necessary knowledge through the search

for external knowledge in the relationship of a firm's value chain [34]. This construct was assessed using nine items adapted from Sambasivan et al. [41]. Second, the technological innovation capability was defined as the step of transforming the knowledge and technology accumulated in the partnership into a new product. Following the operationalization by Zahra and Nielsen [45], we adopted four indicators. Third, alliance proactiveness was defined as the propensity to find strategic alliance opportunities with potentially valuable firms [55]. This construct was assessed using four items adapted from Sarkar et al. [55]. Finally, international performance was assessed using five items from Knight and Cavusgil [18]. The measurement items were all measured using a five-point Likert scale, ranging from 1 (strongly disagree) to 5 (strongly agree).

In addition, this study employed the following several control variables. First, the firm age (measured in years in operation) was controlled according to the possibility of differences in the international performance. Second, the number of employees was controlled, as this could affect the international performance. These two variables were log-transformed to correct for any bias. Third, industry dummies were also used. Fourth, alliance experience can affect the international performance as they can learn strategic alliance management methods, accumulate knowledge, or develop competencies. Accordingly, alliance experience existence (1) and non-existence (0) were created as dummy variables and used for analysis. Finally, differences in degree of Internationalization can affect the international performance. This was controlled by measuring the number of international market entries on a Likert five-point scale, ranging from 1 (one country) to 5 (five or more countries).

## 4. Analysis and Results

### 4.1. Validity and Reliability

To verify the construct validity, we conducted exploratory factor analysis (Table 2). Principal component analysis was used as the analysis method, and analysis was performed using the varimax rotation. As a result, all items were classified into four variables. Additionally, all items were loaded significantly on the corresponding latent construct with acceptable values of standardized factor loading ranging from 0.607 to 0.863. These results indicated sufficient convergent validity. Next, the construct reliability confirmed the internal consistency using Cronbach's alpha. The results indicated that all values exceeded the acceptable threshold of 0.70 [65], ranging from 0.856 (technological innovation capability) to 0.915 (relational capital).

As shown in Table 3, all variables indicate the means, standard deviations, and correlations. The overall correlations between constructs were found to be moderate. However, the correlation value between exploration and exploitation was as high as 0.720, and multicollinearity analysis was performed. Multicollinearity was considered to exist if the variation inflation factor (VIF) was greater than 10 as recommended by Hair et al. [66]. The VIF was found to be 1.066 to 1.703, indicating no multicollinearity between variables.

### 4.2. Hypothesis Testing

In this study, a hierarchical regression analysis was run to test our hypotheses. As reported in Table 4, the relational capital ($\beta = 0.408$, $p < 0.01$) was found to have a significant positive effect on the technological innovation capability in support of Hypothesis 1. We also found that the technological innovation capability had a significant positive effect on the international performance ($\beta = 0.414$, $p < 0.01$), supporting Hypothesis 2.

**Table 2.** Measurement of constructs with multiple items.

| Item | 1 | 2 | 3 | 4 |
|---|---|---|---|---|
| Relational capital 7 | 0.798 | | | |
| Relational capital 6 | 0.783 | | | |
| Relational capital 4 | 0.764 | | | |
| Relational capital 5 | 0.755 | | | |
| Relational capital 8 | 0.752 | | | |
| Relational capital 3 | 0.719 | | | |
| Relational capital 9 | 0.714 | | | |
| Relational capital 1 | 0.612 | | | |
| Relational capital 2 | 0.607 | | | |
| International performance 2 | | 0.818 | | |
| International performance 1 | | 0.793 | | |
| International performance 3 | | 0.788 | | |
| International performance 4 | | 0.782 | | |
| International performance 5 | | 0.746 | | |
| Alliance proactiveness 2 | | | 0.847 | |
| Alliance proactiveness 3 | | | 0.840 | |
| Alliance proactiveness 1 | | | 0.795 | |
| Alliance proactiveness 4 | | | 0.789 | |
| Technological innovation capability 2 | | | | 0.863 |
| Technological innovation capability 3 | | | | 0.845 |
| Technological innovation capability 4 | | | | 0.780 |
| Technological innovation capability 1 | | | | 0.634 |
| Eigen value | 8.524 | 2.700 | 2.106 | 1.792 |
| % of variance | 38.747 | 12.274 | 9.574 | 1.792 |
| Cumulative explained variance (%) | 38.747 | 51.021 | 60.595 | 68.742 |
| Cronbach's alpha | 0.915 | 0.897 | 0.897 | 0.856 |

**Table 3.** Means, standard deviations, and correlations.

| Variable | Mean | SD | 1 | 2 | 3 | 4 | 5 | 6 | 7 | 8 | 9 | 10 | 11 |
|---|---|---|---|---|---|---|---|---|---|---|---|---|---|
| 1. Firm age [1] | 2.87 | 0.618 | 1 | | | | | | | | | | |
| 2. Firm size [1] | 4.87 | 0.628 | 0.014 | 1 | | | | | | | | | |
| 3. Electronics | 0.21 | 0.405 | −0.014 * | 0.137 | 1 | | | | | | | | |
| 4. Textile | 0.11 | 0.312 | 0.128 | −0.230 * | −0.178 * | 1 | | | | | | | |
| 5. Other | 0.11 | 0.312 | 0.097 | −0.030 | −0.178 * | −0.122 | 1 | | | | | | |
| 6. AE | 0.76 | 0.425 | 0.015 | −0.316 * | −0.086 | 0.150 * | 0.020 | 1 | | | | | |
| 7. DI | 1.92 | 1.32 | −0.083 | 0.146 | 0.085 | −0.146 | −0.035 | −0.013 | 1 | | | | |
| 8. RC | 3.53 | 0.609 | −0.006 | −0.001 | −0.019 | −0.101 | 0.131 | −0.009 | 0.098 | 1 | | | |
| 9. TIC | 3.05 | 0.821 | −0.013 | 0.014 | −0.050 | −0.044 | 0.102 | 0.159 * | 0.007 | 0.335 ** | 1 | | |
| 10. AP | 3.09 | 0.711 | −0.083 | 0.113 | −0.002 | −0.225 * | 0.091 | 0.051 | 0.062 | 0.475 ** | 0.350 ** | 1 | |
| 11. IP | 3.34 | 0.616 | −0.072 | −0.084 | −0.096 | 0.013 | −0.012 | 0.232 ** | −0.035 | 0.414 ** | 0.454 ** | 0.349 ** | 1 |

Note: AE = Alliance experience; DI = Degree of internationalization; RC = Relational capital; TIC = Technological innovation capability; AP = Alliance proactiveness; IP = International performance; SD = Standard deviation * $p < 0.05$, ** $p < 0.01$ (two-tailed tests).
[1] Log transformation.

To verify the mediation effect, we implemented bootstrap analysis by using the PROCESS macro in SPSS [67]. For the analysis, 5000 bootstrap samples and a 95% confidence interval were set. As reported in Table 5, the total effect between the relational capital and international performance was found to be significant (β = 0.411, $p < 0.01$). The direct effect of the parameter of technological innovation capability was also significant (β = 0.313, $p < 0.01$). As a result, Hypothesis 3 was adopted, as the direct effect decreased compared to the total effect, indicating that there was a mediating effect.

Next, a hierarchical regression analysis was performed to verify the moderating effect of the alliance proactiveness on the relationship between the technological innovation capability and international performance. To mitigate the potential threat of multicollinearity between the variables, mean-centering was conducted prior to generating the interaction terms [68]. As shown in Table 4, the results showed that the relationship between the technological innovation capability and international performance was significantly mod-

erated by the alliance proactiveness ($\beta = 0.142$, $p < 0.05$). This finding provides support for Hypothesis 4.

**Table 4.** Results of the hierarchical regression analysis.

| Variable | Model 1 | Model 2 | Model 3 | Model 4 |
|---|---|---|---|---|
| Firm age | −0.030 (0.097) | −0.088 (0.069) | −0.072 (0.067) | −0.087 (0.067) |
| Firm size | 0.094 (0.103) | −0.008 (0.073) | −0.030 (0.071) | −0.010 (0.071) |
| Electronics | 0.022 (0.153) | 0.094 (0.110) | 0.089 (0.108) | 0.100 (0.107) |
| Textile | −0.010 (0.234) | 0.047 (0.165) | 0.087 (0.163) | 0.094 (0.161) |
| Other | 0.071 (0.229) | −0.031 (0.163) | −0.044 (0.158) | −0.030 (0.157) |
| Alliance experience | 0.210 ** (0.147) | 0.167 * (0.108) | 0.152 * (0.105) | 0.163 * (0.104) |
| Degree of internationalization | −0.038 (0.046) | −0.046 (0.033) | −0.049 (0.032) | −0.050 (0.031) |
| Relational capital | 0.408 ** (0.099) | | | |
| Technological innovation capability (A) | | 0.414 ** (0.053) | 0.341 ** (0.054) | 0.329 ** (0.054) |
| Alliance proactiveness (B) | | | 0.233 ** (0.062) | 0.264 ** (0.063) |
| Interaction term(A × B) | | | | 0.142 * (0.047) |
| $R^2$ | 0.138 | 0.239 | 0.284 | 0.303 |
| $\Delta R^2$ | - | - | 0.045 | 0.018 |
| F | 3.293 ** | 6.337 ** | 7.057 ** | 6.899 ** |

Note: Standardized coefficients are reported. Standardized errors appear in the parenthesis. Model 1 dependent variable: technological innovation capability. Models 2–4 dependent variable: international performance. * $p < 0.05$, ** $p < 0.01$ (two-tailed tests).

**Table 5.** Mediation analysis with bootstrapping.

| Path | Effect | β | SE | 95% | |
|---|---|---|---|---|---|
| | | | | LLCI | ULCI |
| RC → TIC → IP (H3) | Total effect | 0.411 ** | 0.070 | 0.272 | 0.549 |
| | Direct effect | 0.313 ** | 0.070 | 0.175 | 0.451 |
| | Indirect effect | 0.097 ** | 0.033 | 0.042 | 0.170 |

Note: RC = Relational capital; TIC = Technological innovation capability; IP = International performance; LLCI: lower levels of confidence interval; ULCI: upper levels of confidence interval; ** $p < 0.01$.

## 5. Discussion

### 5.1. Results and Contributions

The purposes of this study were to examine the causal relationship between the relational capital, technological innovation capability, and international performance of SMEs and to verify the moderating effect of alliance proactiveness. To achieve these purposes, research hypotheses were established based on previous research and data were collected by distributing a structured questionnaire for empirical analysis.

The analysis results were as follows. First, we found that the high level of relational capital for SMEs had a positive effect on building the technological innovation capabilities

required for international markets. These results were found to be consistent with Saxenian's [43] research results. Second, the technological innovation capabilities of SMEs were identified as an important factor in increasing the internationalization performance, and the arguments in the study of Zahra and Nielsen's [45] research were also confirmed in this study. Finally, when SMEs had high alliance proactiveness, it appeared that they enhanced their technological innovation capabilities to increase their international performance, which supports Kale and Singh's [64] results.

In international business fields, studies have been conducted for the successful internationalization of firms. In particular, studies focused on the identification of the preceding factors to increase international performance have been mainly conducted. However, most of the studies have focused on multinational corporations centering on large corporations, and the research in the strategic aspect to increase the international performance of SMEs is relatively insufficient.

Accordingly, this study focused on technological innovation capabilities, including external collaboration, as a method to increase the international performance of SMEs from an open innovation perspective. Accordingly, the results of this study emphasize the importance of technological innovation capabilities in order for SMEs to increase their international performance. These results can also be seen as contributing to theoretical expansion as they support an open innovation perspective where technological innovation capabilities are not built independently but need to cooperate with external partners.

### 5.2. Practical Implications

The practical implications are as follows. First, SMEs can build a firm's technological innovation capabilities through relational capital, which represents all the knowledge inherent in relationships with stakeholders, such as customers, suppliers, and competitors. This is because of the quality of relationships that can be formed through communication and trust with external stakeholders, and, through this, high-level knowledge exchange can be obtained to rescue technological innovation capabilities. Therefore, this implies that SMEs must secure the technical knowledge necessary for international markets by forming strong relationships with customers, suppliers, and competitors.

Second, SMEs are limited in performing various activities due to a lack of resources. In particular, the importance of strategic assets is more emphasized in a complex international market environment, for instance the liability of foreignness. In the results of this study, not only the direct effect of technological innovation capability on international performance but also the mediating effect between relational capital and the internationalization performance were verified, and their importance was further emphasized. Therefore, if SMEs reinforce their technological innovation capabilities, they will be able to develop the new ideas and new products necessary for international markets and be able to achieve internationalization results.

Third, SMEs will be able to achieve internationalization by strengthening their technological innovation capabilities through alliance proactiveness. When SMEs conduct new strategic alliances, they have the advantage of being able to access the tacit and formal knowledge of their partners. This means that it is possible to learn about the skills of partners through performing alliance tasks, thereby enhancing the absorptive capacity. Thus, SMEs need to actively seek out opportunities for new alliances to further increase their knowledge range in homogeneous and heterogeneous industries. This suggests that it is necessary to build the capacity to seize the necessary opportunities in international markets.

### 5.3. Limitations and Suggestions for Future Research

Despite these contributions and implications, this research has several limitations that are presented with suggestions for future studies. First, this study estimated causality between variables using only questionnaire data. Therefore, a more rigorous estimation may be possible if research is conducted using longitudinal data in future studies. Second, this study conducted research on the manufacturing industry. However, when considering

the alliance proactiveness of open innovation for international performance, there may be a difference in the service industry. In future studies, it would be of interest to conduct comparative studies divided into manufacturing and service industries.

Fourth, from a knowledge management point of view, looking at the transformation process of knowledge that flows into a firm through relational capital at the individual and team level is meaningful. Therefore, in future studies, multi-level studies considering various variables are necessary. Finally, in spite of this, although this study investigated the relationships among relational capital, technological innovation capital, and international performance in SMEs, we did not consider the dynamic effect of open innovation. Thus, we would like to suggest that future studies consider the dynamic effect of open innovation [69–71].

**Author Contributions:** Conceptualization, J.Y. and D.R.; Methodology, D.R.; Software, J.Y.; Validation, K.H.B.; Formal analysis, K.H.B.; Investigation, D.R.; Resources, K.H.B.; Data curation, K.H.B.; Writing—original draft preparation, J.Y. and D.R.; Writing—review and editing, J.Y.; Supervision, J.Y.; Project administration, J.Y. and K.H.B. All authors have read and agreed to the published version of the manuscript.

**Funding:** This research received no external funding.

**Institutional Review Board Statement:** Not applicable.

**Informed Consent Statement:** Not applicable.

**Data Availability Statement:** Not applicable.

**Acknowledgments:** This work was supported by the 2019 Yeungnam University Research Grant.

**Conflicts of Interest:** The authors declare no conflict of interest.

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
