# Peer review of "Open Innovation with Relational Capital, Technological Innovation Capital, and International Performance in SMEs"

_sustainability, doi:10.3390/su13063418_

Round 1

Reviewer 1 Report

The paper addresses an interesting topic, that is, the Effects of Relational Capital and Technological Innovation Capability on International Performance in SMEs and the Moderating of Alliance Proactiveness,

There are some aspects for improvement that need to be addressed, namely: 

1) Concerning the Abstract:

-Please add description of study novelty as well as more detailed methodology description 

2) Concerning the Introduction and Literature Review:

- I miss some discussion of the concept of Sustainability. Please discuss the importance of your study for this topic.

-I personally believe that the paper is well-grounded in the literature, but the authors must enrich the paper by citing more recent relevant papers. It should be noted that out of 61 references, only 8 are from the last 5 years.

3) Concerning Methodology and Empirical Results:

-Please introduce the concept of Smes you use in the study

-Please present the model figure to be tested in the methodology section. Also, point the hypotheses on figure.

-In Table 4 the authors present the following results: Standardized Structural Estimates of the Structural Model. I do not understand these results, as the use of Structured Equation Modeling (SEM) is never mentioned in the article. Can you please explain?

4) Concerning Discussion and Conclusion:

-In the discussion section the authors must introduce a more dynamic discussion contrasting the paper’s findings with prior works and authors contributions.

Author Response

Dear Reviewer 1. 

Thank you for your kind comments. Please see the attachment.

Reviewer 2 Report

Dear authors, your paper requires several

improvements in order to be accepted.

1) please provide a more detaield analysis of the variables used in your model. A table could favor the comprehension.

2) only few variables are statistically significant. Have you checked the risks related to endogeneity and multicollinearity?

3) relational capital is part of intellectual capital. In literature review you should describe this interlinkage.

4) please extend your discussions with more insights about the managerial and theoretical implications 

Author Response

(The authors gave the same response as above.)

Round 2

Reviewer 1 Report

Well done. Good job.

Author Response

Dear Reviewer 1, 

We(authors) appreciate sincerely your sincere and precious comments. In virtue of these valuable comments, our study could be developed further and be much more improved to be a meaningful research for the academia and the industry.

In virtue of reviewer 1’ kind and most valuable comments, this paper could be much more developed and reformed to be more meaningful to technology society, academia, and entrepreneurs.

Sincerely, we (authors) appreciate about your suggestion and advice to make the better paper, as well as are much grateful for the kindest service and consideration of Editor-in-Chief and the person related Sustainability, and anonymous reviewers. Thank you very much.

Best Wishes,

Authors

Reviewer 2 Report

The paper can be accepted

Author Response

Dear Reviewer 2, 

We(authors) appreciate sincerely your sincere and precious comments. In virtue of these valuable comments, our study could be developed further and be much more improved to be a meaningful research for the academia and the industry.

In virtue of reviewer 1’ kind and most valuable comments, this paper could be much more developed and reformed to be more meaningful to technology society, academia, and entrepreneurs.

Sincerely, we (authors) appreciate about your suggestion and advice to make the better paper, as well as are much grateful for the kindest service and consideration of Editor-in-Chief and the person related Sustainability, and anonymous reviewers. Thank you very much.

Best Wishes,

Authors